# Systematic Generation of Patient-Derived Tumor Models in Pancreatic Cancer

**DOI:** 10.3390/cells8020142

**Published:** 2019-02-10

**Authors:** Karl Roland Ehrenberg, Jianpeng Gao, Felix Oppel, Stephanie Frank, Na Kang, Tim Kindinger, Sebastian M. Dieter, Friederike Herbst, Lino Möhrmann, Taronish D. Dubash, Erik R. Schulz, Hendrik Strakerjahn, Klara M. Giessler, Sarah Weber, Ava Oberlack, Eva-Maria Rief, Oliver Strobel, Frank Bergmann, Felix Lasitschka, Jürgen Weitz, Hanno Glimm, Claudia R. Ball

**Affiliations:** 1Translational Functional Cancer Genomics, National Center for Tumor Diseases (NCT) Heidelberg and German Cancer Research Center (DKFZ), 69120 Heidelberg, Germany; roland.ehrenberg@nct-heidelberg.de (K.R.E.); gjp01891@rjh.com.cn (J.G.); felix.oppel@klinikumbielefeld.de (F.O.); stephanie.frank@nct-heidelberg.de (S.F.); na.kang@nct-heidelberg.de (N.K.); tim.kindinger@nct-heidelberg.de (T.K.); sebastian.dieter@nct-heidelberg.de (S.M.D.); friederike.herbst@nct-heidelberg.de (F.H.); tdubash@mgh.harvard.edu (T.D.D.); erik.schulz@med.uni-heidelberg.de (E.R.S.); strakerjahn@stud.uni-heidelberg.de (H.S.); klara.giessler@gmx.de (K.M.G.); weber.sarah@mail.de (S.W.); ava.oberlack@googlemail.com (A.O.); eva-maria.rief@mayerstiftung.de (E.-M.R.); hanno.glimm@nct-dresden.de (H.G.); 2Department of Medical Oncology, National Center for Tumor Diseases (NCT) Heidelberg, 69120 Heidelberg, Germany; 3German Consortium for Translational Cancer Research (DKTK) Heidelberg, 69120 Heidelberg, Germany; 4Department of Translational Medical Oncology, National Center for Tumor Diseases (NCT) Dresden and German Cancer Research Center (DKFZ), 01309 Dresden, Germany; lino.moehrmann@nct-dresden.de; 5Center for Personalized Oncology, University Hospital Carl Gustav Carus Dresden at TU Dresden, 01307 Dresden, Germany; 6Department of General Surgery, Heidelberg University Hospital, 69120 Heidelberg, Germany; oliver.strobel@med.uni-heidelberg.de; 7Institute of Pathology, Heidelberg University Hospital, 69120 Heidelberg, Germany; frank.bergmann@med.uni-heidelberg.de (F.B.); felix.lasitschka@med.uni-heidelberg.de (F.L.); 8Department of Visceral, Thoracic and Vascular Surgery, University Hospital Carl Gustav Carus Dresden at TU Dresden, 01307 Dresden, Germany; juergen.weitz@uniklinikum-dresden.de; 9German Consortium for Translational Cancer Research (DKTK) Dresden, 01307 Dresden, Germany

**Keywords:** pancreatic cancer, preclinical in vitro model, patient-derived primary culture

## Abstract

In highly aggressive malignancies like pancreatic cancer (PC), patient-derived tumor models can serve as disease-relevant models to understand disease-related biology as well as to guide clinical decision-making. In this study, we describe a two-step protocol allowing systematic establishment of patient-derived primary cultures from PC patient tumors. Initial xenotransplantation of surgically resected patient tumors (n = 134) into immunodeficient mice allows for efficient in vivo expansion of vital tumor cells and successful tumor expansion in 38% of patient tumors (51/134). Expansion xenografts closely recapitulate the histoarchitecture of their matching patients’ primary tumors. Digestion of xenograft tumors and subsequent in vitro cultivation resulted in the successful generation of semi-adherent PC cultures of pure epithelial cell origin in 43.1% of the cases. The established primary cultures include diverse pathological types of PC: Pancreatic ductal adenocarcinoma (86.3%, 19/22), adenosquamous carcinoma (9.1%, 2/22) and ductal adenocarcinoma with oncocytic IPMN (4.5%, 1/22). We here provide a protocol to establish quality-controlled PC patient-derived primary cell cultures from heterogeneous PC patient tumors. In vitro preclinical models provide the basis for the identification and preclinical assessment of novel therapeutic opportunities targeting pancreatic cancer.

## 1. Introduction

Pancreatic cancer (PC) is one of the deadliest malignancies due to its rapid progression, early distant metastasis, late diagnosis and resistance to therapy. It is currently the fourth leading cause of cancer-related deaths in the USA and is projected to be the third leading cause by 2030, surpassing colorectal cancer and breast cancer [1]. So far, the five-year survival rate of PC is approximately 8%, with most patients dying within six months after initial diagnosis [2]. During the past decade, international next-generation sequencing efforts and functional analyses have revealed high levels of inter- and intratumor heterogeneity in multiple malignancies including PC [3,4,5,6]. Recent studies in PC have established tumor cell plasticity and heterogeneity as responsible drivers of progression and differential sensitivity towards chemotherapies [7,8]. Precision medicine approaches aim at tailoring therapy decisions according to the patient’s genetic tumor make-up. However, for a large proportion of patients, treatment recommendations are still sparse and additional strategies are needed to identify and understand patient-specific vulnerabilities. Available standard tumor models like commercially available PC cell lines, cell-line-based xenografts and genetically engineered mouse models (GEMMs) have greatly enhanced the field’s understanding of cellular and pathological processes in PC development and progression. However, defined mouse models harbor a limited repertoire of genetic mutations, and available cell lines mostly do not reflect the full inter- and intratumoral heterogeneity of PC patients [9]. In contrast, patient-derived in vitro and in vivo models established from individual patients directly after surgery of their pancreatic tumors closely reflect the original tumors and facilitate the screening for effective therapeutic approaches or identification of novel vulnerabilities using functional genomics [10,11,12]. For PC, the generation of primary cultures is time-intensive and usually large amounts of viable primary tumor material are required [13]. Moreover, the establishment of primary cell cultures from patient-derived xenograft models has proven to be difficult due to the overgrowth of mouse stromal cells which reduce establishment efficiency [14,15,16]. We here report a 2-step approach allowing the systematic generation of primary pancreatic cancer cell cultures from multiple histological types of pancreatic cancer.

## 2. Materials and Methods

A detailed step-by-step protocol for processing, in vivo expansion and establishment of primary cultures is provided as a resource in the Appendix A.

### 2.1. Purification of Tumor Tissue

All experiments with human material were performed in accordance with the guidelines of the Declaration of Helsinki and were approved by the ethics committee of the Medical Faculty at the University Heidelberg (323/2004, Amendment 03). Informed consent was received from participants before study inclusion. Pieces of tumor tissue were collected from patients undergoing surgery at the Department of Surgery, University Hospital (Heidelberg, Germany) at 4 °C in PBS + 0.1 mg/mL penicillin/streptomycin (PBS/PS). Tumor tissue was minced into small pieces (1–2 mm in diameter), followed by three washings with 20 mL PBS/PS. Tumor pieces were incubated with 20 mL of digestion medium (1× medium 199 (Gibco, Darmstadt, Germany), 2 mg/mL collagenase IV (Invitrogen, Darmstadt, Germany) and 3mM CaCl_2_ (Sigma-Aldrich, München, Germany) at 37 °C for up to 150 min at constant rotation followed by filtering through a 100 µm strainer (BD Biosciences, Heidelberg, Germany). Leftovers on the strainer were further cultivated in vitro.

### 2.2. In Vitro Cultivation of Pancreatic Cancer Cells

Partially digested tumor minces were cultured in Advanced DMEM-F12 medium supplemented with 6 mg/mL d-Glucose, 2% B27-supplement (1×), 2 mM L-glutamine, 5 mM HEPES buffer and 6 μg/mL heparin sodium salt. Fibroblast growth factor (rhFGF-basic, 10 ng/mL, R&D Systems, Wiesbaden, Germany), rhFGF10 (20 ng/mL, R&D Systems, Wiesbaden, Germany) and rhNodal (20 ng/mL, R&D Systems, Wiesbaden, Germany) were added to the culture medium and renewed every 3–4 days. Medium was changed twice per week or when beginning to turn orange. When they reached 80–90% confluency, cells were detached by accutase (PAA) and split 1:1 to 1:10. Cultures were tested for authenticity and contamination, utilizing Multiplex Cell Line Authentication (MCA) and Cell Contamination Test Analyses (McCT; Multiplexion, Heidelberg, Germany).

### 2.3. Laboratory Animals

Male or female immune-deficient NOD.Cg-Prkdc^scid^Il2rg^tm1Wjl^/SzJ (NSG) mice purchased from The Jackson Laboratory (Bar Harbor, Maine, USA) were further expanded in the Centralized Laboratory Animal Facilities of the German Cancer Research Center of Heidelberg. Animals were group-housed in standard individually ventilated cages with wood chip embedding (LTE E-001, ABEDD, Vienna, Austria), nesting material, autoclaved tap water and ad libitum diet (autoclaved mouse/rat housing diet 3437; Provimi Kliba AG, Kaiseraugst, Switzerland). Room temperature and relative humidity were adjusted to 22.0 ± 2.0 °C and 55.0 ± 10.0% respectively, in accordance with Appendix A of the European Convention for the Protection of Vertebrate Animals used for Experimental and Other Scientific Purposes from March 19, 1986. According to the recommendations of the Federation for Laboratory Animal Science Associations (FELASA) all animals were housed under strict specific pathogen-free (SPF) conditions. The light/dark (L/D) cycle was adjusted to 14 h lights on and 10 h lights off with the beginning of the light and dark period set at 6:00 a.m. and 8:00 p.m., respectively. For all experiments eight- to fourteen-week-old female or male NSG mice, weighing 21–30 g, were used. All animal experimentation performed in this study was conducted according to the national guidelines and was reviewed and confirmed by an institutional review board/ethics committee headed by the responsible animal welfare officer. The Regional Authority of Karlsruhe, Germany finally approved the animal experiments in its jurisdiction as the responsible national authority (approval numbers G-76/12 and G233-15).

### 2.4. Xenotransplantation of Pancreatic Cancer Tissue and Cells

Tumor pieces of 1–5 mm size of minced pancreatic cancer tissue (or 2 × 10^6^ to 7 × 10^6^ purified cells in the case of serial xenotransplantation) were transplanted subcutaneously into eight- to fourteen-week-old female or male NSG mice anesthetized by 1.75% isoflurane in the breathing air. Carprofen (4 µg per gram body weight) was administered as a painkiller. Mice were checked daily and sacrificed by cervical dislocation when tumor volume reached 1.5 cm^3^ or 26 weeks after transplantation.

### 2.5. Histological Inspection of Pancreatic Cancer Tissue

A representative proportion of patient and xenografted tumor covering multiple compartments of the tumor (usually approximately 0.2–0.3 cm^3^) was routinely checked by a senior pathologist who specialized in pancreas-specific pathology. Therefore, samples were fixed with 10% formalin (Sigma-Aldrich, München, Germany), dehydrated by an ascending ethanol series and embedded in paraffin wax (Merck, Darmstadt, Germany). Ten-micrometer tissue slices were stained by hematoxylin and eosin. 

### 2.6. Indirect Immunofluorescence (IF) Staining

Cells were grown on 20 mm coverslips (Th. Geyer, Renningen, Germany) and fixed in ice cold 4% paraformaldehyde (Carl Roth, Karlsruhe, Germany) in PBS (Invitrogen, Darmstadt, Germany). Cells were blocked in PBS + 0.1% BSA (Sigma-Aldrich) and permeabilized by PBST solution (0.1% Triton X-100 (AppliChem, Darmstadt, Germany) and 0.1% sodium citrate (in PBS) at 4 °C. DNA was stained by Hoechst (Invitrogen, Darmstadt, Germany). The actin cytoskeleton was visualized by phalloidin-PF647 (Promokine, Heidelberg, Germany). Marker-stained cells were incubated for 1 to 16 h in a wet chamber employing the following primary antibodies (dilution 1:100): Rabbit anti-human KRT7 (product code ab53123, Abcam, Cambridge, England); mouse anti-human KLF4 (clone 56CT5.1.6, Abgent, Oxfordshire, England); mouse anti-human THY1 (clone AS02, Dianova, Hamburg, Germany); mouse anti-human KRT7 (clone OV-TL 12/30, Dako system, Hamburg, Germany), mouse anti-human αSMA (clone 1A4, 1:200, Sigma-Aldrich), mouse anti-human Nestin (clone 10c2, Santa Cruz, Heidelberg, Germany), mouse anti-human Desmin (clone D33, Santa Cruz, Heidelberg, Germany) and mouse anti-human vimentin (clone V9, Santa Cruz, Heidelberg, Germany). Afterwards, cells were incubated with secondary antibodies (dilution 1:200) for 1 h: Donkey anti-mouse IgG-Dylight649 (code number 715-495-150, Jackson IR, Suffolk, England); donkey anti-rabbit IgG-Dylight549 (code number 711-505-152, Jackson IR, Suffolk, England); goat anti-mouse IgG-PF555 (catalogue number PC-PK-PF555-AK-M1, Promokine, Heidelberg, Germany); goat anti-rabbit IgG-PF488 (catalogue number PK-PF488P-AK-R1, Promokine, Heidelberg, Germany).

### 2.7. Flow Cytometry Analysis

Cells were suspended in Hank’s Balanced Salt Solution (HBSS) (Sigma-Aldrich, München, Germany) supplemented with 2% fetal bovine serum (FBS) (Pan Biotechnology, Aidenbach, Germany) (HF buffer) and distributed into FACS tubes with up to 1 × 10^6^ cells per tube. TOTO^®^-3 (Thermo Fisher, Schwerte, Germany) for dead cell indication according to the manufacturer’s protocol was applied. An antibody-detecting specific cell-surface marker was diluted in the HF buffer according to manufacturer instruction and resuspended cells were incubated at 4 °C for 30 min. Human EpCam (clone EBA-1, dilution 1:20, BD Biosciences, Heidelberg, Germany), Human CD45 (clone HI-30, dilution 1:100, BD Bioscience, Heidelberg, Germany) and murine H2kd (clone SF1-1.1, dilution 1:50, BD Bioscience, Heidelberg, Germany) were used for phenotyping. Flow cytometry was performed using LSRII or AriaII (BD Bioscience, Heidelberg, Germany) and FACS Diva software (Version 6.1.3) (BD Bioscience, Heidelberg, Germany).

## 3. Results

We processed a total of 134 surgical specimens from patients diagnosed with pancreatic cancer to generate patient-derived in vitro and in vivo tumor models. Initially, tumor pieces were directly cultured in vitro in analogy to the outgrowth method, which was first described to isolate pancreatic stellate cells from non-malignant fibrotic pancreatic tissue (Figure 1A) [6,17]. Therefore, fresh tissue material was minced into small pieces and washed in PBS followed by enzymatic digestion. Filtering of the digestion mix resulted in larger, partially digested tumor pieces that were further cultivated under serum-free conditions in advanced DMEM-F12 medium with supplements and beneficial cytokines (FGF-basic, FGF-10 and Nodal). Under these conditions, cancer cells grew out and epithelial colonies formed, accompanied by cells of fibroblast or myofibroblast morphology (Figure 1B). However, when outgrowth cultures were further cultivated in vitro, epithelial cells were overgrown by stroma cells after passaging (9/9) (Figure 1C). Xenotransplantation of cultivated primary cell cultures of two patients with at least partial epithelial morphology resulted in xenograft tumors (4/4) with pancreatic adenocarcinoma histology in immunocompromised NSG mice (Figure 1D). The mesenchymal origin of fibroblast-like cells was validated by detection of stroma markers Vimentin, THY1 and α-smooth-muscle actin, whereas epithelial cell colonies could be distinguished from stroma cells by the expression of the pancreatic duct marker Cytokeratin 7 (Figure 1E–G). Stroma-dominant cultures were characterized by Vimentin and THY1 positivity as well as high CD44, but low EpCam and CD24 and no detectable CD133 expression (Appendix A). Even though stromal cultures could be further cultivated in vitro, they completely lacked tumorigenicity and none of 11 NSG mice transplanted with 10^5^ to 10^6^ cells from 2 different patients formed tumors.

### 3.1. In Vivo Expansion of Primary Tumors from Pancreatic Cancer Patients

As tumor material available for patient-derived tumor model generation mostly did not exceed 1.0 g of weight (average of 0.81 g, range 0.1 g to 2.6 g), we included a xenotransplantation expansion step to amplify primary tumor material prior to in vitro cultivation. The tissue was cut in small pieces and subcutaneously transplanted into an immunocompromised NSG mouse (Figure 2A). In cases where available tumor tissue weight exceeded 1.0 g, two mice were transplanted per patient. Whenever feasible, a representative tumor piece was fixed for histological validation. A detailed step-by-step protocol for the processing of the PC tissue, serial xenografting and primary culture establishment is provided in the Appendix A.

Of all 134 xenotransplanted patient tumors, 54 were successfully engrafted in NSG mice (54/134, 40.3%; Appendix A). Following harvesting of the primary xenografts, tumors were digested into single cell suspensions (1.5 × 10^6^ to 7.3 × 10^7^ cells per tumor with an average of 2.1 × 10^7^ cells) and purified cells were serially re-transplanted into two parallel NSG mice (0.2 × 10^6^ to 7.3 × 10^6^ purified cells transplanted per NSG mouse). Whenever both parallel mice engrafted, they were sacrificed simultaneously (average time to tumor harvest 120 days ± 63 days) to enable re-unification of tumor cells derived from paired tumors following purification (89%, 48/54). In summary, of the 54 patient tumors which successfully engrafted in mice, a total of 26 were transplanted for one mouse generation, 8 for two consecutive mouse generations and additional 20 patient tumors were serially transplanted for three consecutive mouse generations.

Histological analysis of those original patients’ tumors which had successfully formed xenograft tumors showed that the majority (44/54, 81.5%) were pancreatic ductal adenocarcinoma (PDAC). According to the Union for International Cancer Control (UICC) staging system, 66.7% (36/54) of patients were in stage IIB at the time of diagnosis. We did not observe a correlation of engraftment efficiency to the weight of the tumor piece received for processing or transplanted cell number (Figure 2B). Furthermore, the tumor stage was not predictive for engraftment (Figure 2C) or the tumor growth time in serial transplantation (Table 1). Patient-specific histology was conserved for all expansion xenografts (Figure 2D). Expansion xenografts tended to harbor slightly reduced stromal cell content together with a more solid pattern of tumor growth (31,6%, 6/19) or a more pronounced mucinous appearance (21,1%, 4/19) compared to the respective primary patient tumor. Of note, from all 54 xenografted patient tumors, 6% (3/54) showed outgrowth of human CD45+ hematopoietic cells in xenografts, presumably triggered by Epstein-Barr Virus (EBV) transformed B-cells as reported before by us and others [18,19]. CD45+ cells were detected at the transplantation site (Appendix A) as well as at the metastatic sites including liver and lymph node (Appendix A), whereas EpCam positive human epithelial cancer cells were limited to a small subpopulation of cells exclusively at the site of xenotransplantation (Appendix A). These three xenografts were not further processed for establishing primary cell cultures.

### 3.2. Establishment of PC Patient-Derived In Vitro Models

A total number of 51 successfully xenografted epithelial tumors were subjected to semi-adherent in vitro cultivation to establish PC patient-derived primary culture models. In brief, following partial digestion of the expansion xenograft tumors into small tumor pieces, these were subsequently cultivated in serum-free cancer stem cell (CSCN) medium supplemented with (rhFGF-basic), rhFGF10 and rhNodal [6,17]. Primary human pancreatic cancer cells grew out from tumor pieces and accompanying murine fibroblasts were gradually outcompeted by cancer cells during subsequent passaging. Primary patient cultures formed semi-adherent colonies with tight cell-to-cell contacts and proliferated robustly over more than 20 cell passages in vitro.

In total, 22 primary pancreatic cancer cell cultures were established (22/51; 43.1%). To confirm the epithelial origin of established primary cultures, expression of the human epithelial marker EpCam, murine cell marker H2kd and blood cell marker CD45 were analyzed by flow cytometry. Among established cultures, 78.6% primary cultures (22/28) were strongly positive for EpCam (range 84%–99.9%, average 96.5%), confirming their origin from human epithelial cells (Figure 3A, Appendix A). However, 6/28 cell cultures (21.4%) were dominated by cells of murine origin as demonstrated by strong positivity (66–100%) for mH2kD staining (Figure 3A) even after three to five rounds of passaging. These murine-contaminated cultures were excluded from further analyses. Upon xenotransplantation, all 22 established primary patient-derived epithelial cultures reliably formed tumors following subcutaneous injection of 10^4^ to 10^7^ cells.

Histological analysis of culture-derived xenograft tumors revealed pancreatic ductal adenocarcinoma histology in >75% of xenografts (Figure 3B), whereas the remaining xenografts were classified as adenosquamous carcinomas. Importantly, the histology of both the xenograft tumor derived from established primary cell culture and its matching patient’s tumor showed consistent tumor histology (Figure 3C)**.** Notably, even complex tumor differentiation, like oncocytic type intraductal papillary mucinous neoplasm (IPMN) mixed with dominant proportion of invasive PDAC, was retained in xenografts derived from primary cultures. As observed in expansion xenografts, primary cell culture-derived xenografts tend to exhibit less stroma and increased proportion of mucinous components with goblet cells or more solid tumor growth pattern compared to the respective patient’s histology.

## 4. Discussion

Despite extensive preclinical and clinical research in pancreatic cancer, treatment options for PC are still limited, and median survival after diagnosis is less than a year [1]. Patient-derived tumor models as preclinical models that faithfully recapitulate the complexity and heterogeneity of pancreatic cancer are considered ideal preclinical models for the identification of novel target structures using large scale screening approaches, as well as for the implementation of personalized treatment approaches into clinical practice [20].

In recent years, different protocols for the establishment of patient-derived in vitro models (e.g., patient-derived cell cultures (PDC) and xenografts (PDX)) as well as pancreatic cancer organoids have been reported [13,21,22,23,24]. Despite their great impact, model generation can be limited by technical or practical drawbacks. For example, serial xenotransplantation as a prerequisite for testing treatment strategies in xenografted mouse cohorts for individual tumor patients takes up to several months to establish [25]. Consequently, screening or drug testing approaches in PDX models can be time- and labor-intensive, which limits their potential for prospective personalized therapy approaches [16,24]. Organoid models of pancreatic cancer, on the other hand, have been shown to allow highly efficient seeding in vitro [13,26], maintain clonal heterogeneity, reflect plasticity and allow comparison of normal and malignant pancreatic cells. However, PC organoid protocols still lack standardization, and the high costs of daily maintenance and specialized conditions so far hamper their widespread use [27].

Our protocol allows reasonably efficient generation of patient-derived cancer models which maintain the histological structure of matching patients’ tumor and reflect the pathological diversity of pancreatic cancer. Of note, one caveat in successful model generation is the potential loss of primary cultures by excessive proliferation of non-cancerous human cell types. First, human fibroblasts can eventually outcompete epithelial cancer cells in initial cultures in vitro. Including a preceding in vivo xenograft expansion step prior to culture generation replaced human fibroblast during xenograft growth by murine stromal cells and increased efficiency of tumor culture establishment. Second, immortalized patient-derived B-cells within the tumor tissue may harbor a strong proliferative capacity and can outcompete cancer cells in primary culture [18,19]. Regular flow cytometry analyses to identify CD45 positive cells enables prospective monitoring of tumor cell content of proliferating primary cultures and should therefore regularly used to assess culture quality.

Histologically, pancreatic cancer is characterized by extensive fibrosis, with stroma outnumbering cancerous counterparts [28]. Therefore, pancreatic cancer tissue from surgical resection contains a limited amount of viable cancerous cells. This limits the efficiency of cancer cell engraftment during direct in vitro cultivation of human PC tissue, as stromal cells under these circumstances can rapidly overgrow the epithelial cancer cells. Our strategy of including an initial xenotransplantation step prior to in vitro cultivation expands the malignant cells while the human stroma perishes [29,30].

Multiple factors including the viability of the primary tumor sample, the sterility of the tumor handling and the genetic mutation status (such as Smad4 loss) determine successful engraftment of patient tumor tissue following xenotransplantation into immunodeficient mice [24,31,32]. In line with previous studies [33], our results demonstrate that the engraftment of primary tumor material is independent from either tumor stage or the weight of the transplanted tumor, and even enables successful generation of patient-derived in vitro models from PDAC as well as adenosquamous carcinomas and a partial oncocytic differentiated carcinoma. Moreover, histoarchitecture of primary cell culture-derived xenografts closely resembles their matching primary patients’ tumors. To maximize engraftment rates, focused selection of tumor pieces with highly malignant cell content as estimated by microscopic validation may be implemented in future settings.

Of note, in xenograft models of human gastrointestinal cancers, genetic subclones can be highly dynamic in gastrointestinal cancers [34,35]. Different environmental conditions, like the mouse environment in xenotransplantation, or potential subclonal differences due to regional sampling of tissue for model generation and sequencing, may be responsible for these observations. Despite such subclonal dynamics, the majority of driver alterations within gastrointestinal tumors remain clonal [36,37]. Moreover, in colorectal cancer, functional heterogeneity within the tumor-initiating cell compartment is independent from genetic subclone heterogeneity [35], further underlining the utility of such patient-derived tumor models as disease-relevant tumor models in preclinical experimental research.

During the growth of xenotransplanted human tumor, murine stromal cells are recruited to replace the human tumor-associated stroma in vivo and interact with cancer cells to promote tumor growth [16,38,39]. Therefore, cells purified from the xenografted tumors are composed of both human cancer epithelial cells and murine stromal cells. Eventually, cell cultures established from xenografts can be contaminated by murine stroma as observed in our study and reported by others [14,40,41]. Novel approaches like repeated trypsinization combined with epithelial-enhanced media, immunomagnetic mouse cell depletion (MCD) kit, and EpCAM-based immunomagnetic positive selection to eliminate murine stroma contamination may be further tested to see whether they reduce the rate of cultures with overwhelming murine cell contamination even further [14,41,42].

In summary, we here provide a systematic two-step protocol enabling the systematic establishment of heterogeneous patient-derived tumor models of multiple pancreatic cancer subtypes. These models may serve as a platform for understanding mechanistic dependencies in future translational research settings.

## Figures and Tables

**Figure 1 cells-08-00142-f001:**
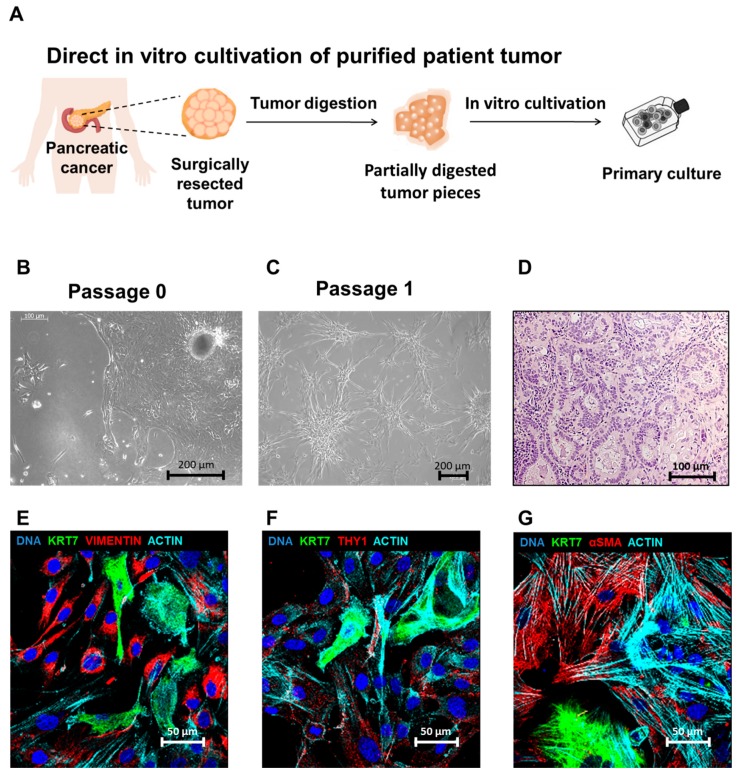
Cultivation of pancreatic tumor cells directly after resection of patient tumors. (**A**) Following surgery, pancreatic cancer tissue was minced and enzymatically digested. The partially digested tumor pieces were plated in cell culture flasks for culture establishment. (**B**) Primary tumor cells grew out from the tumor pieces and grew as epithelial colonies. (**C**) Following passaging, the epithelial cells gradually were lost and were overgrown by fibroblast-like cells. (**D**) Xenograft tumor derived from the cell culture established directly from surgically resected cancer tissue displayed adenocarcinoma histology. (**E**–**G**) Outgrowth cultures established directly from surgically resected tumor pieces expressed the pancreatic duct marker Cytokeratin 7 (KRT7), mesenchymal stroma markers vimentin (**E**), THY1 (**F**) and α-smooth-muscle actin (αSMA, red) together with actin (cyan) (**G**). (**B**, **C**) Representative data of PC42 is shown, (**D**–**G**): representative data of PC2 are shown.

**Figure 2 cells-08-00142-f002:**
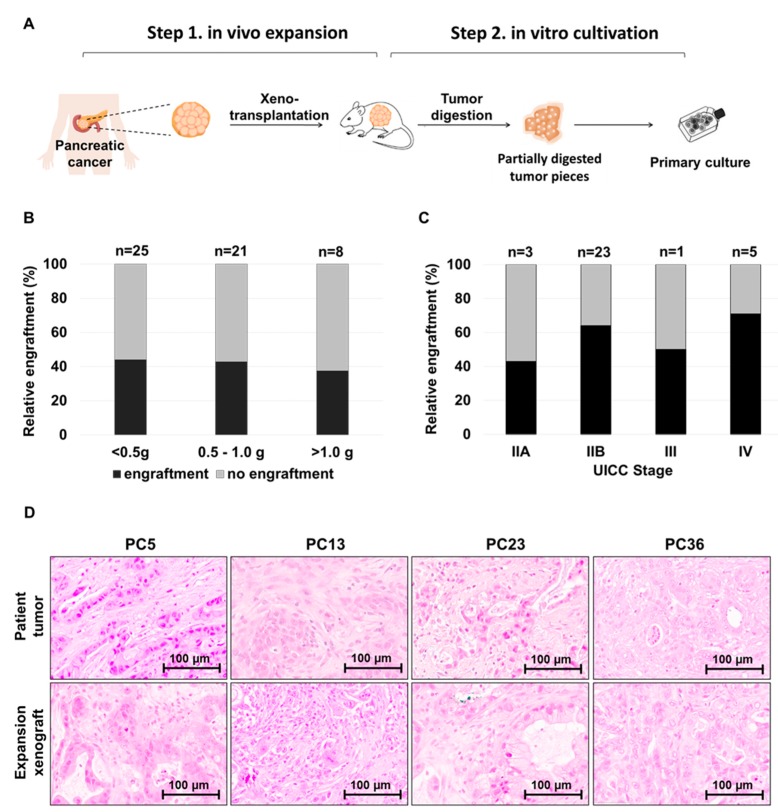
Establishment of primary pancreatic tumor cell cultures following in vivo expansion. (**A**) Surgically resected pancreatic cancer tissue was directly transplanted in NSG mice for in vivo expansion of viable tumor cells (average time to tumor harvest 120 days ± 63 days). Following xenograft formation, tumors were harvested, processed and used for culture establishment and serial re-transplantation. (**B**, **C**) Engraftment of patient-derived tumor pieces was independent of the weight (**B**) of transplanted tumor and the UICC Stage (**C**) of the matching patient; n = numbers of transplantation cases (each case represents the xenotransplantation of one patient’s tumor). (**D**) Histology of original patients’ tumors is recapitulated by matching expansion xenograft tumors. PC5 and PC 36: Ductal adenocarcinoma; PC13: Adenosquamous carcinoma; PC23: Mucinous ductal adenocarcinoma.

**Figure 3 cells-08-00142-f003:**
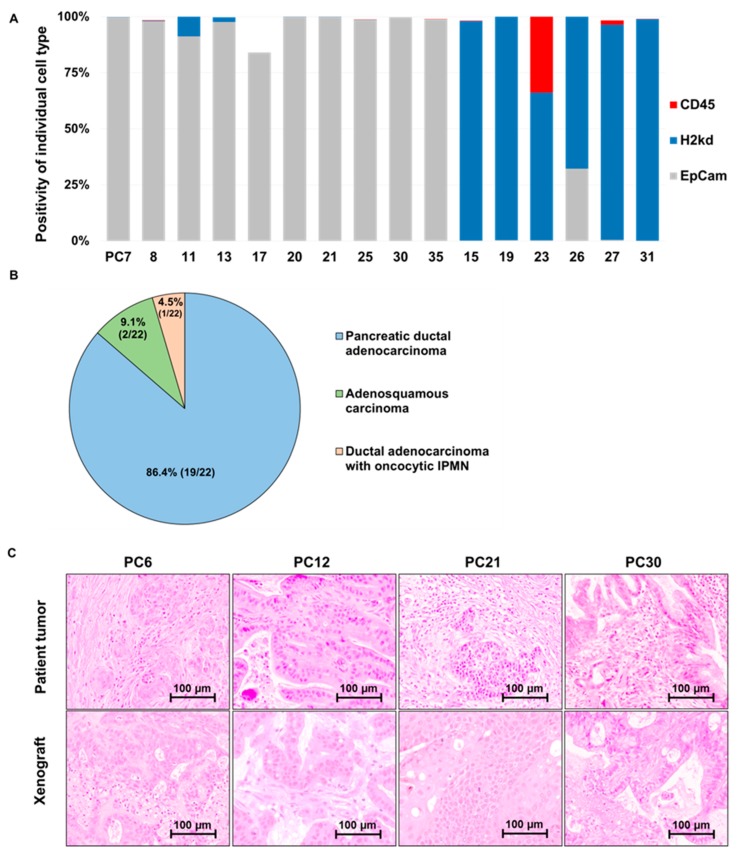
Characterization of established patient-derived primary cultures. (**A**) FACS analysis demonstrates epithelial identity of successfully established patient-derived in vitro culture as well as 6 cultures dominated with murine stromal cells as indicated by prevalence of murine H2Kd expressing cells. (**B**) Primary cell cultures (n = 22) could be established from human pancreatic cancers with different histology PC: Ductal adenocarcinoma (n = 19), adenosquamous carcinoma (n = 2), and ductal adenocarcinoma with oncocytic IPMN: Ductal adenocarcinoma with oncocytic intraductal papillary mucinous neoplasm (n = 1). (**C**) Upon xenotransplantation of established primary patient derived cultures, the xenograft tumors closely resemble the respective patient tumor histology.

**Table 1 cells-08-00142-t001:** Average days (mean) of in vivo tumor growth for xenograft expansion until tumor harvest.

UICC Stage	n	1st	SD	2nd	SD	3rd	SD
IIA	3	186	± 138	93	± 59.6	48	± 19.8
IIB	23	103	± 43.4	64	± 16.2	58	± 14.2
III	1	127		37		21	
IV	5	146	± 67.5	81	± 22.2	51	± 14.6

1st: first generation xenograft, 2nd: second generation xenograft; 3rd: third generation xenograft; UICC: Union for International Cancer Control; SD: standard deviation.

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
