# Peer review of "Systematic Generation of Patient-Derived Tumor Models in Pancreatic Cancer"

_cells, 2019, doi:10.3390/cells8020142_

Round 1
Reviewer 1 Report
The study by Ehrenberg et al describes efforts to generate patient derived cultures for better understanding of pancreatic cancer (PnCa) progression and for use as personalized disease model to identify therapies that most likely would benefit the patient. Despite several advances in genomic sequencing and development of targeted therapies for PnCa, the overall survival has not improved significantly and new models that reflect the disease heterogeneity and could be used to select therapies and for drug discovery are urgently needed. The current manuscript describes a two-step protocol including the initial expansion of surgically resected PnCa into the subcutaneous region of immunodeficient mice, followed by primary xenograft cell culture to generate culture from diverse subtypes of PnCa. The protocol was used on 143 surgical specimen and achieved a reasonable 43.1% success in established cultures from xenograft tissues. The authors suggest that their protocol allows the systematic generation of PnCa cultures that mimic the original tumors from which they were derived and could be used to personalize therapies. There are a number of points that must be clarified and some overstatement that should be minimized in the current manuscript.
Specific comments:
1. The authors must present the timeline for the two-step protocol and indicate the time needed to generate the xenograft derived growth followed by the culture in the main manuscript, to avoid misleading the reader, in particular when the studies are compared to slice cultures and pancreatic cancer organoids (PDOs) that could be accomplished in a few weeks.
2. Without sequencing data comparing the derived cultures and xenograft to the original tumors, the authors can’t claim that their xenograft derived cell cultures maintain heterogeneity and are matching the patient tumors.
3. In light of the finding that Patient-derived xenografts undergo murine-specific tumor evolution, including those from PnCa, (Ben-David et al., Nature Genetics 2017), the authors must discuss the potential for mouse-specific clonal selection of human tumors of their engrafted and cultured cells with their protocol.
4. Line 296, the conclusion that this two-step protocol “combines efficient model generation with comparably modest costs while maintaining the histological structure of matching patient’s tumor” is not supported by the data presented.
5. Line 166, the authors need to describe which media and culture conditions have been used for outgrowth cultures to be grown by stroma cells after passaging. Did the authors use pancreatic epithelial supportive culture conditions?
6. The established patient derived cell cultures in general and at least in Fig. 3C should be further characterized by pancreatic specific IHC.
7. Results section, line 240; “all 51 xenografted epithelial tumors”.. Previous section indicated there were 54 engraftments, please clarify.
8. The quality of the images in Fig. 1e-g should be better. DNA staining is blue in the nuclei, why is it overlapping with the SMA staining in e?
Author Response
Revision of manuscript: cells-417731 - Minor Revision
We thank the reviewers for their constructive comments. Please find below a detailed point-by-point response addressing all of the Reviewers’ recommendations.
Reviewer 1:
The study by Ehrenberg et al describes efforts to generate patient derived cultures for better understanding of pancreatic cancer (PnCa) progression and for use as personalized disease model to identify therapies that most likely would benefit the patient. Despite several advances in genomic sequencing and development of targeted therapies for PnCa, the overall survival has not improved significantly and new models that reflect the disease heterogeneity and could be used to select therapies and for drug discovery are urgently needed. The current manuscript describes a two-step protocol including the initial expansion of surgically resected PnCa into the subcutaneous region of immunodeficient mice, followed by primary xenograft cell culture to generate culture from diverse subtypes of PnCa. The protocol was used on 143 surgical specimen and achieved a reasonable 43.1% success in established cultures from xenograft tissues. The authors suggest that their protocol allows the systematic generation of PnCa cultures that mimic the original tumors from which they were derived and could be used to personalize therapies. There are a number of points that must be clarified and some overstatement that should be minimized in the current manuscript.
We have revised our manuscript to implement all suggested modifications.
Specific comments:
1. The authors must present the timeline for the two-step protocol and indicate the time needed to generate the xenograft derived growth followed by the culture in the main manuscript, to avoid misleading the reader, in particular when the studies are compared to slice cultures and pancreatic cancer organoids (PDOs) that could be accomplished in a few weeks.
In vivo expansion of primary patient tumors requires 120 days on average (±63 days), subsequent establishment of primary cell culture varies in a patient specific manner from few days up to several weeks. We modified the manuscript to specify these details (Lines 207, 231).
2. Without sequencing data comparing the derived cultures and xenograft to the original tumors, the authors can’t claim that their xenograft derived cell cultures maintain heterogeneity and are matching the patient tumors.
We agree with the reviewer and apologize for the unprecise description of this aspect. In our manuscript, we describe the generation of primary cell cultures from pancreatic cancer patient tissue from heterogeneous subtypes of human pancreatic cancers. Established cultures form serially transplantable xenograft tumors which closely mirror the original patients’ tumor histology. We therefore conclude that our culture protocol enables the generation of tumor models from heterogeneous patient derived pancreatic tumors and may be useful to further investigate disease biology in experimental and translational research settings. In this respect, such cultures may serve as models to further evaluate the clonal dynamics and clonal heterogeneity in patient tumors and derived tumor models. We modified the manuscript accordingly (Lines 332-340).
3. In light of the finding that Patient-derived xenografts undergo murine-specific tumor evolution, including those from PnCa, (Ben-David et al., Nature Genetics 2017), the authors must discuss the potential for mouse-specific clonal selection of human tumors of their engrafted and cultured cells with their protocol.
The reviewer correctly points out that potential subclonal selection needs to be considered in experimental settings involving murine xenotransplantation of human tumor tissue. In line with Ben-David et al, we have shown before in colorectal cancer that multiple co-existing genomic subclones within individual patient derived xenografts can fuel long term tumor-initiation (Gießler et al 2017). Importantly, despite potential subclonal selection processes triggered by different environmental conditions (e.g. mouse) and/or potential subclonal differences due to regional sampling of tissue for model generation and sequencing, these and other multi-region sequencing studies (Jesinghaus et al., 2015, Sottoriva et al., 2015) showed that the majority of tested driver mutations were clonal. These findings support the utility of such patient derived tumor models as disease relevant tumor models in preclinical experimental research. In our study, we consequently pooled xenografted tumor tissue from parallel mice prior to serial re-transplantation to at least partially correct for potential clonal selection in individual mice. Indeed, we show that xenografted tumors closely mirror the original patients’ tumor histology. We have modified the manuscript to discuss these points in more detail (Lines 332-340).
4. Line 296, the conclusion that this two-step protocol “combines efficient model generation with comparably modest costs while maintaining the histological structure of matching patient’s tumor” is not supported by the data presented.
We have modified our conclusion as suggested by the reviewer (Lines 303-305).
5. Line 166, the authors need to describe which media and culture conditions have been used for outgrowth cultures to be grown by stroma cells after passaging. Did the authors use pancreatic epithelial supportive culture conditions?
For serial passaging of outgrowth cultures we used similar culture conditions as for culture initiation. In brief, cells were cultured in Advanced DMEM-F12 medium supplemented with D-Glucose, B27-supplement, L-glutamine, HEPES buffer and Heparin sodium salt and pancreatic cancer cell supportive cytokines were added (10 ng/ml rhFGF-basic, 20 ng/ml rhFGF10) and 20 ng/ml rhNodal. As these conditions do not selectively support growth of human pancreatic cancer cells only, but also were allowed proliferation of human stromal cells, stromal cells grew out and dominated the cultures with time. We have modified to manuscript to make this point more clear (Lines 168-169).
6. The established patient derived cell cultures in general and at least in Fig. 3C should be further characterized by pancreatic specific IHC.
Within the manuscript, we characterize and define successfully established cultures by their strong positivity for the epithelial marker EpCam and the lack of expression of murine H2kd as well as human CD45. In addition, we demonstrate that the cultures give rise to xenografts closely resembling the patient specific tumor histology. Together, these data conclusively support pancreatic cancer derived origin of the cultures. We agree that IHC would enable an even broader characterization of the cultures. However, the cost and time consumption of these efforts would add only limited additional information. We therefore feel that such analyses are beyond the scope of the current manuscript and need to be addressed separately.
7. Results section, line 240; “all 51 xenografted epithelial tumors”.. Previous section indicated there were 54 engraftments, please clarify.
Among all 54 xenografted patient tumors, three cases show outgrowth of human CD45+ hematopoietic cells in xenografts. Therefore, these three xenografts were not further processed for in vitro cultivation. We adjusted our manuscript to make this point more clear (Line 246).
8. The quality of the images in Fig. 1e-g should be better. DNA staining is blue in the nuclei, why is it overlapping with the SMA staining in e?
We thank the reviewer for bringing this point to our attention and apologize for the limited figure quality. For re-submission we provide a corrected version of the image at improved quality. Here, we added the information that co-staining for actin was done using Phalloidin-PF647 (cyan color). The actin/aSMA signal partially overlaps with the nuclear staining, as actin fibers spanning to the cytoplasm can compress the nucleus or appear in the nucleus as an artefact of the staining/imaging process. We modified the manuscript accordingly (Lines 182, 192).

Reviewer 2 Report
Ehrenberg et al in this manuscript have very clearly presented the methodology: 2-step approach allowing the systematic generation of primary pancreatic cancer cell cultures from multiple histological types of pancreatic. The authors have in a precise manner shown how the methodology has been developed in a systematic manner. Overall the paper has several relevant pieces of information that would add value to the current set of information available in the literature. Development of such invivo and invitro models would lead to the availability of more model systems that could be used to elucidate the assessment of novel therapeutic opportunities targeting pancreatic cancer. This would help in finding out better treatment modalities.
Overall the data is well presented and neatly written. The manuscript may be accepted for publication. It just needs a few minor details.
1) In the methodology "Laboratory Animals" what age group mice did the authors use for their study?
2) In the discussion, the authors may need to discuss/address if there are any caveats they observed and how they resolved them.
Author Response
Revision of manuscript: cells-417731 - Minor Revision
We thank the reviewers for their constructive comments. Please find below a detailed point-by-point response addressing all of the Reviewers’ recommendations.
Reviewer 2:
Ehrenberg et al in this manuscript have very clearly presented the methodology: 2-step approach allowing the systematic generation of primary pancreatic cancer cell cultures from multiple histological types of pancreatic. The authors have in a precise manner shown how the methodology has been developed in a systematic manner. Overall the paper has several relevant pieces of information that would add value to the current set of information available in the literature. Development of such invivo and invitro models would lead to the availability of more model systems that could be used to elucidate the assessment of novel therapeutic opportunities targeting pancreatic cancer. This would help in finding out better treatment modalities.
Overall the data is well presented and neatly written. The manuscript may be accepted for publication. It just needs a few minor details.
We thank the Reviewer for his positive comments.
1) In the methodology "Laboratory Animals" what age group mice did the authors use for their study?
For xenotransplantation experiments, we used eight to fourteen weeks old (male or female) NSG mice (Lines 116, 117, 125).
2) In the discussion, the authors may need to discuss/address if there are any caveats they observed and how they resolved them.
Within the new version of the manuscript, we have extended the discussion of caveats in model generation by describing in more detail the potential loss of primary cultures by excessive proliferation of non-cancerous human cell types. First, human fibroblasts can eventually outcompete epithelial cancer cells in initial cultures in vitro. Including a preceding in vivo xenograft expansion step prior to culture generation led to loss of human fibroblast during xenograft growth due to replacement by murine stromal cells and allows increased efficiency of tumor culture establishment. Second, immortalized patients derived B-cells within the tumor tissue may harbor strong proliferative capacity and can outcompete cancer cells in primary culture. Regular flow cytometry analyses to identify CD45 positive cells proved to be highly efficient to closely monitor tumor cell content of proliferating primary cultures and were therefore included as regular quality control. We have modified the manuscript to further highlight these aspects (Lines 303-314).

Reviewer 3 Report
Line 97, 98, 100 missing manufacturer for FGF , nodal and PAA
Line 127: does it mean a pathologist was checking the implanted tumors “clinically”? if so would be better to rephrase, explaining the meaning of “representative proportion” and what exactly the pathologist was checking (ex: size).
Figure 1: Possibly write magnification scale inside the figure
Line 198 and 199, how long after ingraft in mice were the xenograft tumors harvested?
Do you think mixing cells from subsequent transplanted mice would generate multiple clones in the culture and possibly be a confounding factor impairing culture purity?
Figure 2d: magnification written on the scale bar segment inside the picture would be preferable. I d suggest a higher magnification in this case. Are the patients tumor of different histology? Would be better to write histology in figure legend or inside the figure
Figure 3c: better to have magnification on scale bar. It s suggested a higher magnification to better show histology
Not clear what is the difference in term of efficacy and reproducibility of histology, between the method in figure 1 a and the one in figure 2 a. Could that be stated more clearly?
Supplementary methods
Step 4 of purification of pancreatic cancer: when is the supernatant discarder? there is no previous centrifugation step..
Step 5: how many cells are transplanted?
About “Establishment and daily handling of pancreatic cancer primary cultures”
1: the undigested tumor tissue handling is described, what about the digested part?
Author Response
Revision of manuscript: cells-417731 - Minor Revision
We thank the reviewers for their constructive comments. Please find below a detailed point-by-point response addressing all of the Reviewers’ recommendations.
Reviewer 3:
Line 97, 98, 100 missing manufacturer for FGF, nodal and PAA
We have included the manufacturer information into the material and methods section (Line 97, 98, 99).
Line 127: does it mean a pathologist was checking the implanted tumors “clinically”? if so would be better to rephrase, explaining the meaning of “representative proportion” and what exactly the pathologist was checking (ex: size).
We apologize for not sufficiently clarifying this issue in the manuscript. A senior pathologist specialized in pancreas specific pathology routinely evaluated and compared patient and xenograft tumor histologies following HE staining. Therefore, representative tumor slices were generated from paraffin embedded tumor or xenograft pieces covering all macroscopic distinguishable compartments of the tumor. We adjusted the manuscript accordingly (Lines130-132).
Figure 1: Possibly write magnification scale inside the figure
We included magnification scales inside the figure as suggested.
Line 198 and 199, how long after ingraft in mice were the xenograft tumors harvested?
Time to tumor formation during in vivo expansion of primary patient tumors in NSG mice requires 120 days on average (±63 days). We modified the results text to make this point more clear (Lines 207, 231).
Do you think mixing cells from subsequent transplanted mice would generate multiple clones in the culture and possibly be a confounding factor impairing culture purity?
Indeed genetic subclone kinetics have been described within in patient derived serial xenograft models by us and others (Gießler, Ben-David). As in parallel mice from the same patient different subclones may grow and and eventually dominate, we seeked to maintain a broader subclone heterogeneity by pooling xenografted tumor tissue from parallel mice prior to serial re-transplantation. We have modified the manuscript to discuss this aspect in more detail (Lines 332-340).
Figure 2d: magnification written on the scale bar segment inside the picture would be preferable. I d suggest a higher magnification in this case. Are the patients tumor of different histology? Would be better to write histology in figure legend or inside the figure
We included images with higher magnification and magnification scales inside the figure. Patient tumors shown are diagnosed as ductal adenocarcinoma (PC5 and PC36), Adenosquamous carcinoma (PC13) and mucinous ductal adenocarcinoma (PC23). We added this information into the figure legend (Lines 237, 238).
Figure 3c: better to have magnification on scale bar. It s suggested a higher magnification to better show histology
We included images with higher magnification and magnification scales inside the figure as suggested.
Not clear what is the difference in term of efficacy and reproducibility of histology, between the method in figure 1 a and the one in figure 2 a. Could that be stated more clearly?
Figure 1a depicts our efforts to generate cultures directly from patient tissue. However, due to overgrowing stromal cells, we were not able to generate stable cultures (efficacy 0%). We therefore modified our strategy by including an in vivo xenotransplantation step in immundeficient mice which allowed expansion of vital tumor cells for culture generated. Figure 2 shows that using this two-step method (Fig. 2a), primary cultures were successfully established from the expansion xenografts in 22 of 51 samples (efficacy 43%). We modified the manuscript to further highlight this point (Lines 303-314).
Step 4 of purification of pancreatic cancer: when is the supernatant discarder? there is no previous centrifugation step
We thank the reviewer for his comment and included the description of the centrifugation step into the protocol.
Step 5: how many cells are transplanted?
For serial xenotransplantation, a total of 0.2 x106 to 7 x106 cells were transplanted per NSG mouse. We included this information into the manuscript (Lines 124 and Supplementary methods).
About “Establishment and daily handling of pancreatic cancer primary cultures”
1: the undigested tumor tissue handling is described, what about the digested part?
Purified tumor cells can be either directly subjected to experimental settings or stored in liquid nitrogen for later use. We modified the protocol accordingly.
